# DICER1 Syndrome: A Multicenter Surgical Experience and Systematic Review

**DOI:** 10.3390/cancers15143681

**Published:** 2023-07-19

**Authors:** Claudio Spinelli, Marco Ghionzoli, Linda Idrissi Sahli, Carla Guglielmo, Silvia Frascella, Silvia Romano, Carlo Ferrari, Fabrizio Gennari, Giovanni Conzo, Riccardo Morganti, Luigi De Napoli, Lucia Quaglietta, Lucia De Martino, Stefania Picariello, Anna Grandone, Caterina Luongo, Antonella Gambale, Armando Patrizio, Poupak Fallahi, Alessandro Antonelli, Silvia Martina Ferrari

**Affiliations:** 1Division of Pediatric Surgery, Department of Surgical Pathology, University of Pisa, 56126 Pisa, Italy; 2Departmental Section of Medical Genetics, S. Chiara Hospital, 56126 Pisa, Italy; 3Division of Pediatric Surgery, Regina Margherita Hospital, 10126 Turin, Italy; 4Division of General and Oncologic Surgery—Department of Cardiothoracic Sciences, University of Campania “Luigi Vanvitelli”, Via Pansini 1, 80131 Naples, Italy; 5Section of Statistics, University Hospital of Pisa, 56124 Pisa, Italy; 6Division of Endocrine Surgery, Department of Surgical, Medical, Molecular Pathology and of the Critic Area, University of Pisa, 56126 Pisa, Italy; 7Neuro-Oncology Unit, Department of Paediatric Oncology, Santobono-Pausilipon Children’s Hospital, 80123 Naples, Italy; 8Department of Woman, Child of General and Specialized Surgery, University of Campania “L. Vanvitelli”, 80138 Naples, Italy; 9CEINGE Advanced Biotechnology, 80131 Naples, Italy; 10Integrated Care Department of Laboratory Medicine, Unit of Medical Genetics, Federico II Hospital, 80131 Naples, Italy; 11Department of Emergency Medicine, Azienda Ospedaliero-Universitaria Pisana, 56126 Pisa, Italy; 12Department of Translational Research and New Technologies in Medicine and Surgery, University of Pisa, 56126 Pisa, Italy; 13Department of Surgical, Medical and Molecular Pathology and Critical Area, University of Pisa, 56126 Pisa, Italy; 14Department of Clinical and Experimental Medicine, University of Pisa, 56126 Pisa, Italy

**Keywords:** DICER1, neoplasms, genetic mutation, surgery, children

## Abstract

**Simple Summary:**

DICER1 syndrome is a rare genetic disorder that predisposes patients to the development of malignant and non-malignant diseases. Currently, DICER1 syndrome diagnosis still occurs late, usually following surgical operations, affecting patients’ outcomes, especially for further neoplasms, which are entailed in this syndrome. For this reason, herein we present a nationwide multicenter report of DICER1 syndrome in the pediatric population, with the prospective aim of enhancing post-surgical surveillance. Moreover, a comprehensive literature review of DICER1 cases, including case reports and multicenter studies published from 1996 to June 2022, was performed. Eventually, the retrieved data from the literature were compared with the data emerging from our cohort of patients. The early identification of patients affected by this mutation would allow a timely screening program, thus avoiding diagnostic delays.

**Abstract:**

DICER1 syndrome is a rare genetic disorder that predisposes patients to the development of malignant and non-malignant diseases. Presently, DICER1 syndrome diagnosis still occurs late, usually following surgical operations, affecting patients’ outcomes, especially for further neoplasms, which are entailed in this syndrome. For this reason, herein we present a multicenter report of DICER1 syndrome, with the prospective aim of enhancing post-surgical surveillance. A cohort of seven patients was collected among the surgical registries of Pediatric Surgery at the University of Pisa with the General and Oncologic Surgery of Federico II, University of Naples, and the Pediatric Surgery, Regina Margherita Hospital, University of Turin. In each case, the following data were analyzed: sex, age at diagnosis, age at first surgery, clinical features, familial, genetic investigations, and follow-up. A comprehensive literature review of DICER1 cases, including case reports and multicenter studies published from 1996 to June 2022, was performed. Eventually, the retrieved data from the literature were compared with the data emerging from our cohort of patients.

## 1. Introduction

DICER1 syndrome, or “familial tumor predisposition syndrome of pleuropulmonary blastoma”, is a rare genetic disorder that predisposes patients to the development of malignant and non-malignant diseases [1]. Individuals with DICER1 syndrome are at increased risk of developing: pleuropulmonary blastoma (PPB), which is the most common neoplasm associated with DICER1; among thyroid disorders, multinodular goiter (MNG) and differentiated thyroid cancers (DTC); cystic nephroma and, less commonly, Wilms’ tumor or anaplastic kidney sarcoma; ovarian tumors such as Sertoli–Leydig cell type; and embryonic rhabdomyosarcoma [2]. Children with DICER1 syndrome may rarely develop: brain tumors, such as pineoblastoma, nasal chondromesenchymal hamartoma, and medulloepithelioma of the ciliary body [2]. The estimated prevalence of DICER1 pathogenic variants in the general population is 1:10,600 and due to the heterogeneous clinical features and rarity of this syndrome, its diagnosis remains a challenge for clinicians [3]. In the oncological population, the prevalence of this disease is estimated at 1:4600 [4,5]. The first symptoms appear in the first two decades of life; there are no significant differences in sex or ethnic groups [3,6]. The DICER1 gene is located on chromosome 14q32.13 and encodes for a protein of approximately 200 kDa. It is an endoribonuclease of the RNase III family involved in the production and maturation process of most microRNAs (miRNAs), which are ~22-nucleotide small noncoding RNAs, known to have a key role in the post-transcriptional regulation of mRNA [5,7]. The deregulation of miRNAs has a pro-oncogenic effect: the overexpression of one miRNA can act to inhibit the protein translation of a tumor suppressor gene, while the downregulation of another miRNA can increase the protein level of an oncogene [8,9,10,11,12]. miRNAs are generated from the nascent primary miRNA (pri-miRNA) transcripts through two consecutive cleavage events. The pri-miRNA is processed by DROSHA in the nucleus, which releases a hairpin-shaped precursor (pre-miRNA). Pre-miRNAs move from the nucleus to cytoplasm by exportin 5 (XPO5) and are cleaved by DICER. The resulting small RNA duplex is loaded onto the Argonaute (AGO) protein, which retains only one strand of mature miRNA and removes the other one. The miRNA-loaded AGO associates with other cofactors and constitutes the effector complex called the RNA-induced silencing complex (RISC). The miRISC induces the decay of mRNA and translational suppression by interacting with the complementary sequences in the 3′-untranslated region (3′-UTR) of target gene mRNA. The miRNAs target most mRNA, and in this way, they have important regulatory roles in different physiological and developmental processes. Overall, miRNA-mediated gene expression control is critical for cell response to oxidative stress, hypoxia, and DNA damage, and for this reason, it is involved in human diseases such as cancer [13] (Figure 1). Presently, DICER1 syndrome diagnosis still occurs late, usually following surgical operations, affecting patients’ outcomes, especially for further neoplasms, which are entailed in this syndrome. For this reason, herein we present a multicenter report of DICER1 syndrome in the Italian pediatric population based on the collaboration of several Pediatric Surgery facilities, thus contributing to the diagnosis of this rare syndrome with the prospective aim of enhancing post-surgical care in order to tackle any other subsequent oncologic manifestation that may occur in these patients.

## 2. Materials and Methods

A cohort of seven patients was collected among the surgical registries of Pediatric Surgery at the University of Pisa with the General and Oncologic Surgery of Federico II, University of Naples, and the Pediatric Surgery, Regina Margherita Hospital, University of Turin. In each case, the following data were analyzed retrospectively: sex, age at diagnosis, age at first surgery, clinical features, familial, genetic investigations, and follow-up.

The search for genetic variants in the coding regions of the causative genes of oncological predisposition syndromes was carried out by target-Next Generation Sequencing (tNGS) analysis of a multi-gene panel including 154 genes implicated in the onset of the syndrome. The filtering of the gene variants was carried out through a dedicated pipeline, considering the variants of the DICER1 gene. The data obtained by t-NGS were confirmed on the patient’s DNA by amplification with specially designed primers and subsequent Sanger sequencing [11].

The follow-up program for patients who tested positive was set up as follows: a chest CT scan and renal ultrasound as a baseline, which was then repeated after 2.5 years; then a chest X-ray every 6 months until age 8 years; and then every 12 months until age 12 years. Starting from the age of 8 years, ultrasounds and blood tests for the thyroid gland were performed; however, if the patient had prior radiation therapy for PPB, ultrasound was performed every 6 months for at least 5 years. Pelvic ultrasound was performed every 1–2 years from infancy to adulthood [2]. 

In this study, a comprehensive literature review of DICER1 cases using PubMed (http://www.ncbi.nlm.nih.gov/PubMed (accessed on 5 July 2023)) offered by the National Library of Medicine, including case reports and multicenter studies published from 1996 to June 2022, was performed. “DICER1” “children” “surgery” “genetic” were used as Medical subject heading (Mesh) terms. Eventually, the retrieved data from the literature were compared with the data emerging from our cohort of patients. 

For data analysis, the categorical data were described by absolute and relative (%) frequency, and the continuous data were summarized by mean and range. To compare percentages and means, z-test for two proportions and t-test were performed, respectively. Significance was fixed at 0.05, and all analyses were carried out with SPSS v.28 technology.

## 3. Results

The results of our study are given in Table 1. The results from the literature review are in Table 2, and the comparison between our study and the literature is in Table 3.

We analyzed seven patients, including three females and four males, with an average age of 6.6 years (range: 2–12 years). The average age at first surgery was 1.7 years (range: 40 days–3 years), while the average age at DICER1 syndrome diagnosis was 2 years (range: 3 months–4 years). We highlighted these two categories, namely the average age at DICER1 syndrome diagnosis and the age at first surgery, in order to highlight that this type of disease is often diagnosed following the first surgery. 

In our cohort of patients, DICER1 diagnosis occurred on average 5.4 months following the first operation, some of which led to the diagnosis of a second neoplasm. The PPB diagnosis occurred in four cases (57%) and was the most common neoplastic manifestation of the syndrome. All patients were treated with lobectomy via a thoracotomy and neoadjuvant/adjuvant chemotherapy; the average age at surgery was 1.6 years (range: 2 months–3 years). Only one case of PPB had already been diagnosed with DICER1 syndrome. Renal involvement was found in four cases (57%). Patients affected by cancer had an average age of 2.25 years (range: 1.5–3 years). Both cystic nephroblastoma (CN) and anaplastic renal sarcoma of the embryonic type were treated with nephrectomy and followed up. The other two cases involved one with asymmetrical kidneys at the age of 2 years and one with multiple bilateral renal cysts at the age of 3 months. None of the four cases had received a DICER1 syndrome diagnosis when renal involvement was detected.

Thyroid nodules were present in two out of seven patients (28%); a thyroid adenoma (TA) and a DTC. The younger patient (4 years) underwent total thyroidectomy, which resulted in a DTC with capsular infiltration, whereas the older patient (11 years) underwent lobectomy. In such cases, DICER1 syndrome diagnosis had already been confirmed, and the follow-up allowed a timely diagnosis. Multinodular goiter (MNG) was diagnosed in one out of seven patients (14%), at the age of 4 years. 

Neurodevelopmental disorders were diagnosed in two cases, both at the age of 2 years. A patient was diagnosed with speech delay, while another was diagnosed with neuropsychiatric delay with a poorly organized motor pattern. Neither of them had previously been diagnosed with DICER1 syndrome. Central Nervous System (CNS) involvement was found in three patients (43%). The average age was 3 years (range: 2–4 years); two of these had already been diagnosed with DICER1 syndrome. In this case, the cerebral electrical activity on electroencephalogram (EEG) appeared diffusely slowed down and misconfigured, while on brain magnetic resonance imaging (MRI), the cerebrospinal fluid (CSF) spaces appeared diffusely prominent, with periventricular gliosis and megacisterna magna. In another case, a pineal cyst was found in brain MRI. 

No malignancies were documented in the latter two cases. The third case was diagnosed with left ocular malignant teratoid medulloepithelioma, which was surgically removed. 

Glow syndrome was diagnosed in a patient at the age of 2 years. The patient was reported to have asymmetrical lower limbs and a subleveled pelvis of about 5 mm on the left. A wrist X-ray showed bone age approximately 1 year and 1 month older than chronological age. Weight and head circumference were >97° pc, with dysmorphisms (prominent forehead, epicanthus, converging strabismus to the right eye, retrognathia, sunken nasal root, bulbous nose, flared and anteverted nostrils, sole and bilaterally incomplete palmar fold, bilaterally retroposed thumb, thoracic and shoulder asymmetry, 4th and 5th toes bilateral clinodactyly, and mild left lower limb hemihypertrophy). This phenotype added to nephroblastoma has been identified as Glow syndrome with somatic DICER1 mutations. 

Embryonic rhabdomyosarcoma (ERMS) presented in a case at the age of 11 months (14.28%), localized on the left forefoot; the patient underwent surgical excision. The genetic diagnosis of DICER1 syndrome came two years after surgery. This patient developed multiple inguinocrural lymph nodes and bilateral lung metastases after two years after the first surgery and eventually died four years after the first operation. 

Taking familial factors into account, three out of seven patients (43%) had a positive family history of DICER1; in all cases, fathers were involved. More specifically, in the first case, namely Case 1, his father was diagnosed with glaucoma at the age of 18. During his second decade of life, he developed toxic MNG and hyperparathyroidism. Finally, he was diagnosed with non-small cell lung adenocarcinoma at the age of 44 years with a positive DICER1 test. In a second patient, namely Case 5, her father was diagnosed with MNG at the age of 9 years and was DICER1-positive. In the last case, namely Case 6, his father was documented positive for DICER1 syndrome during the screening test, although he never developed any symptoms. Siblings were involved in two cases: Case 1, a brother was diagnosed with cystic nephroma at the age of 1 year with DICER1 syndrome, while in Case 5, a one-year-older brother presented DICER1 syndrome with a single-loculated upper polar cyst of the right kidney, later defined as a simple dysplastic cyst. Mutations in DICER1 gene were assessed via genetic tests; blood sample were also retrieved on parents and siblings, as reported in Table 1. 

The follow-up of our patients lasted 3.5 years (range: 1–8 years). To date, four patients are disease-free (57%), one developed a pineal cyst still under surveillance (14%), one patient submitted to surgery for PPB had a persistent lung cystic lesion under surveillance, whereas one patient with embryonal rhabdomiosarcoma (RMS) (14%), died after 2 years from surgery. 

As concerning the literature, most reviews are from the USA, Canada, and UK, with, respectively, 37, 36, and 22 reports. A minor contribution comes from Germany, Italy, China, Japan, Sweden, Spain, France, Turkey, Denmark and Thailand, while no reports are available from Russia, Australia or South America (Figure 2). The distribution of age at diagnosis in the literature review can be found in Figure 3. 

## 4. Discussion

In the present paper, we report a multicenter study of DICER1 syndrome in the pediatric population, with the aim of increasing global awareness about this disease in order to improve clinical outcomes through the timely diagnosis of any neoplastic event. 

Approximately 87% of germline mutations have autosomal dominant inheritance with incomplete penetrance, while 13% are de novo mutations [12]. As reported by a study carried out on a group of subjects with PPB and a DICER1 mutation, up to a fifth of patients have a de novo mutation, and given the variable penetrance of this mutation, patients may have either parents or siblings with silent clinical phenotype [14]. DICER1 syndrome leads to the development of tumors following Knudson’s two-hit theory: the second somatic mutation alters how DICER1 processes miRNAs in some cells [9]. The dysfunctional protein DICER1 produces an abnormal mix of miRNAs. These somatic missense variations have been identified in nearly all DICER1-associated cancers; they are typically found at five “hotspot” codons in the RNaseIIIb domain (E1705, D1709, G1809, D1810, E1813) and alter the cleavage capacity of the protein [1,3,4]. Rarer mosaicisms have also been observed for missense variants that predispose to a more severe phenotype, albeit without the DICER1 germline mutation [6]. 

Concerning the epidemiological features of DICER1 syndrome, the sex distribution demonstrated, in accordance with our cohort, a male-to-female ratio favoring the female sex (62% versus 37%) [1,14,15,16,17,18,19,20,21,22,23,24,25,26,27,28,29,30,31,32,33,34,35,36,37,38,39,40,41,42,43,44,45,46,47,48,49,50,51,52,53,54,55,56,57,58,59,60,61,62,63,64,65,66,67,68,69,70,71,72,73,74,75]. As far as age at first diagnosis of neoplasm is concerned, similarly to our previous report, we present a case of a 4-month-old infant affected by PPB with full-blown genetics [1,14,15,16,17,18,19,20,21,22,23,24,25,26,27,28,29,30,31,32,33,34,35,36,37,38,39,40,41,42,43,44,45,46,47,48,49,50,51,52,53,54,55,56,57,58,59,60,61,62,63,64,65,66,67,68,69,70,71,72,73,74,75]. Lack of deep knowledge of the syndrome can explain the gap between age at first neoplasm and age at DICER1 syndrome diagnosis, as demonstrated in Figure 4. 

It could also be due to a more timely diagnosis, thanks to the genetic screening protocols more recently applied to subjects affected by PPB or multiple clinical manifestations related to DICER1 syndrome. PPB is confirmed as the most frequent clinical manifestation of the syndrome, as it accounts for about half of the cases both in our study and in the references (57% vs. 41%). As highlighted by Gonzàlez et al. [5], PPB can initially occur as multiple lung cysts evolving subsequently into sarchoma. CN is the second most common clinical manifestation in our cases, while it is reported in the literature [76,77] as the third most common neoplasm (42% vs. 14%) and presents around 4 years of age as a unilateral and well-delimited cystic renal mass, similarly to our cases, except one where the tumor developed earlier at 2.5 years. Anaplastic sarcoma of the kidney (ASK) occurred in 14% of our patients versus 1.5% in the reviewed manuscripts and usually presents as monolateral and might occur simultaneously with CN [15]. DTC is reported as 5%, while it was present in 14% of our cases. 

A screening of DICER1 mutations in pediatric DTC as well as thyroid screening ultrasound is recommended for school children with DICER1, which may be followed up every five years or more frequently for those patients under chemotherapy, as children under high-dose chemotherapy for PPB are more prone to develop DTC [11,78,81]. The relationship between MNG and DICER1 syndrome in our cohort showed a similar ratio to previous experiences (28% vs. 34%) [11,82]. Study performed by De Kock et al. [11] highlights how the frequency of MNG in families in which the DICER1 mutation segregates is notoriously higher in the literature than in DTC, occurring in carriers up to 20%. DICER1 guarantees the correct expression of numerous neurocortical genes and contributes to the correct neuronal transmission, suggesting a role not only in oncogenesis but also in nervous system development [5,81,83]. Interestingly enough, CNS alterations were found in about half of our patients, while such reports in the literature are scarce: these are mainly pituitary blastoma, macrocephaly, and spinal hamartomas [16,20]. The correlation between medulloepithelioma and DICER1 syndrome has been found only once through genetic testing and as evidenced medulloepithelioma associated with DICER1 syndrome can be either teratoid or non-teratoid [79]. Glow syndrome occurred in 14% versus 2% in the literature. In Glow syndrome, a gain of function in the PI3K/AKT/mTOR pathway has been described [78,84]. RMS occurrence in our study is 14% versus 16% in literature: the presentation in our case was at the level of the distal extremity of the lower limb, whereas the prevalent localization is reported at the genitourinary level, such as the cervix in females and the prostate in males [21,22,23]. 

Familial features were not found in 57% of our patients versus 29% of the literature. Germline mutations can occur anywhere in the DICER1 gene, causing loss of function; most carriers with a germline DICER1 variant may live a healthy life, although burdened with an increased risk of malignancy [10]. DICER1 tumor suppressor gene is a master regulator of miRNA processing and may be associated with somatic mutations or gene deletions of the remaining allele. The DICER1 protein deficit is interfering with miRNA output, therefore interfering with post-transcriptional gene regulation, leading to neoplasm development [80]. Eighty-eight DICER1 mutations were described for PPB; while most of these are in the regions encoding defined domains, such as helicase domains, the Dicer dimerization domain (DDD), the Piwi/Argonaute, Zwille (PAZ) domain, the RNase III domains, and the double-stranded RNA-binding domain, other mutations reside outside these domains [1].

In our patients, Case 1, variant c.919_932 of the CGTGCCGTATTGGT (p.Arg307Serfs*9) of the DICER1 gene was found in heterozygosity: the subject was found to be a carrier of a germline mutation affecting the DICER1 gene, which consists in the deletion of 14 nucleotide bases in position 919_932 and which involves the sliding of the reading frame with the consequent introduction of a stop codon and interruption of the protein. This protein will be structurally shorter and functionally less active. This variant has not been found in the reviewed literature. According to the American College of Medical Genetics (ACMG) [85], the variant c.[2339delC];[-] found in Case 2 is diagnostic for DICER1 syndrome. Not even this variant was found in the data available in the reviewed literature. Variant c.574-1G>A of Case 3 has not been observed in studies on healthy controls or in databases of causative variants, but the hypothesis according to which it may involve alteration of the splicing, representing a nonsense mutation, is founded. Given the biological characteristics of this alteration and the correspondence between the lesion (embryonic renal sarcoma) and the observed variant, this variant may be causative. In Case 4, we found a mutation already reported in the literature in 2019 [24]. In Case 5, testing revealed the c.1376+G>T (rs886037670-CS118387) mutation in a heterozygous state affecting the DICER1 gene. The mutation in question is thought to cause DICER1 syndrome. This mutation has been found twice in the literature [25,26]. Case 6 tested positive, with a mutation already found in the literature [27]. Case 7 reported a mutation also found in recent literature [27]. 

Overall survival rate was 85.7% at a three-year follow-up from DICER1 diagnosis, versus 92.9% reported in the literature. Among our cases, only an infant affected by ERMS died at the age of five after a three-year follow-up, while seven out of the 132 patients (7%) died in previous reports: three for PPB2 [27,28,86], one for pituitary blastoma [16], one for SLCT [29], one for ovarian tumors of the stroma and sex cords [30], one for RMS [31].

## 5. Conclusions

Currently, DICER1 syndrome remains a diagnostic challenge because symptoms and signs can be unspecific and can have significant differences from one patient to another. Its manifestations can be of malignant or non-malignant nature and can occur exclusively or combined in each patient differently. However, it is important not to underestimate or overlook other atypical associated manifestations, as early identification of patients affected by this mutation would allow a timely screening program, thus avoiding diagnostic delays. Concerning genetic diagnosis, there are still gaps regarding the causative variants, as the same are not described in the reports. For such rare syndrome, an international database including both clinical and genetical details of these patients is still warranted. 

## Figures and Tables

**Figure 1 cancers-15-03681-f001:**
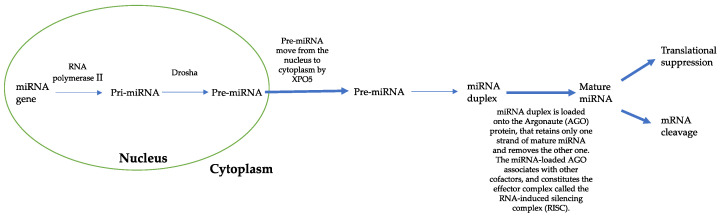
Illustration of miRNA biogenesis dysregulation in cancer.

**Figure 2 cancers-15-03681-f002:**
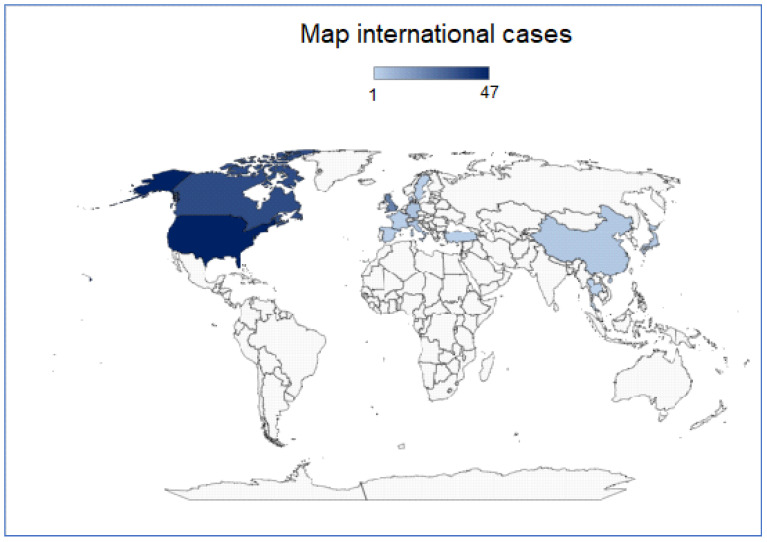
Map of international cases—*Map of the international distribution of DICER1 syndrome cases* [1,1,17,18,19,20,22,23,24,25,26,27,28,29,30,31,32,33,34,35,36,37,38,39,40,41,42,43,44,45,46,47,48,49,50,51,52,53,54,55,56,57,58,59,60,61,62,63,64,65,66,67,68,69,70,71,72,73,74,75,76,77,78,79,80]. [Bing technology, ©Australian Bureau of Statistics, GeoNames, Geospatial Data Edit, Microsoft, Navinfo, OpenStreetMap, TomTom, Wikipedia].

**Figure 3 cancers-15-03681-f003:**
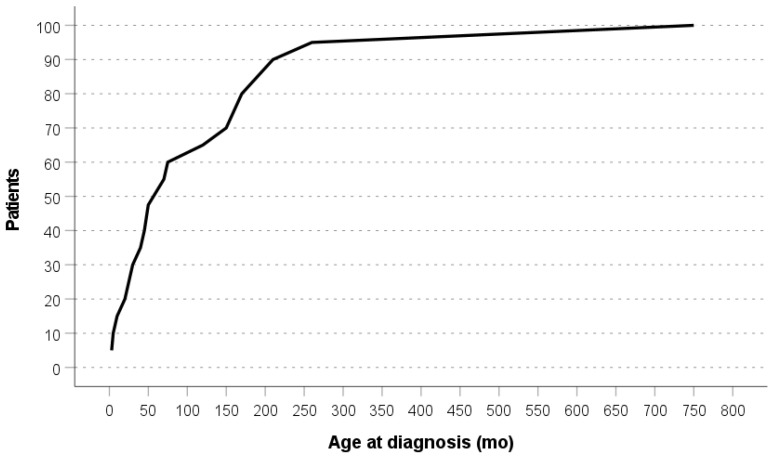
Age distribution at DICER1 syndrome diagnosis in the literature review [1,1,17,18,19,20,22,23,24,25,26,27,28,29,30,31,32,33,34,35,36,37,38,39,40,41,42,43,44,45,46,47,48,49,50,51,52,53,54,55,56,57,58,59,60,61,62,63,64,65,66,67,68,69,70,71,72,73,74,75,76,78,79,80].

**Figure 4 cancers-15-03681-f004:**
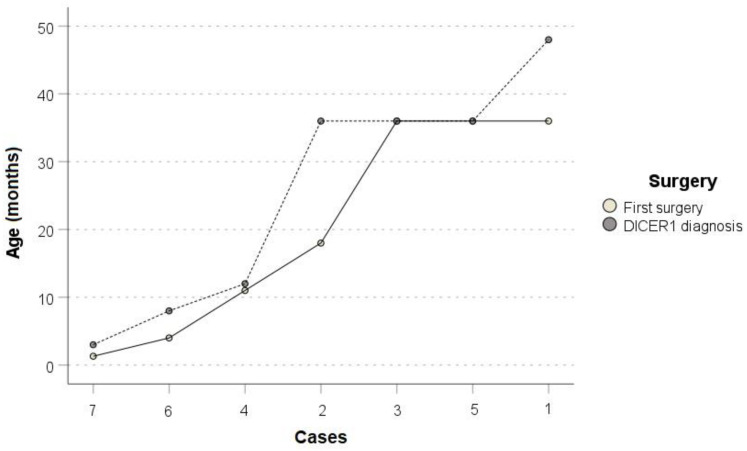
On a marked line chart, age at first surgery (white circles—single line) and age distribution at DICER1 diagnosis (grey circles—dashed line) are reported for institutional cases.

**Table 1 cancers-15-03681-t001:** Case description according to sex, age at surgery, age at DICER1 syndrome diagnosis, clinical manifestations, and genetics.

	Sex	Age at First Surgery (mo)	Age at Diagnosis of DICER1 Syndrome (mo)	Clinical Manifestations	Genetic Variation	Familial	Outcome Status
Case 1	F	36	48	PPB, NG, TA	c.919_932 of CGTGCCGTATTGGT (p.Arg307Serfs*9)	Yes (Father, Brother)	Alive
Case 2	M	18	36	GLOW Syndrome, CN	c.[2339delC]; [-]	No	Alive
Case 3	M	36	36	ASK, MNG, DTC	c.574-1G>A	No	Alive
Case 4	M	11	12	ERMS	c.1069G>A	No	Dead
Case 5	F	36	36	CN, PPB	c.1376+G>T (rs886037670-CS118387)	Yes (Father, Brother)	Alive
Case 6	F	4	8	PPB	c.5465A>T p.D1822V	Yes (Father)	Alive
Case 7	M	1.3	3	CBME, CN, PPB	c.3288_3289insTTTC	No	Alive

Sarcoma of the kidney (ASK); medulloepithelioma of the ciliary body (CBME); cystic nephroblastoma (CN); differentiated thyroid cancers (DTC); embryonic rhabdomyosarcoma (ERMS); nodular goiter (NG); pleuropulmonary blastoma (PPB); glow syndrome (S.GLOW); thyroid adenoma (TA).

**Table 2 cancers-15-03681-t002:** Review of DICER1 cases present in the literature [1,14,15,16,17,18,19,20,21,22,23,24,25,26,27,28,29,30,31,32,33,34,35,36,37,38,39,40,41,42,43,44,45,46,47,48,49,50,51,52,53,54,55,56,57,58,59,60,61,62,63,64,65,66,67,68,69,70,71,72,73,74,75].

Author	Year	Sex	Age at Diagnosis	Clinical Manifestations	Genetic Mutation	Familial	Outcome Status
Hill et al. [21]	2009	3 F 7 M	0.5 0.9 N/A N/A N/A N/A N/A N/A N/A N/A	PPB + cERMS	c.1910dupA c.2268_2271delTTTG c.1507G>T c.2245_2248dupTACC c.2247C>A c.1684_1685delAT c.2392dupA c.2830C>T c.3540C>A c.4748T>G	3 No 7 Yes	1 Dead 9 Alive
Slade et al. [27]	2011	4 F 1 M 14 N/A	1.5 0.8 4 3 7 0.9 4.2 3 4 1.8 1.3 1.5 3 17 12 6 8 13 32	11 PPB, CBME 2CN, 3 SLCT dx, sx, PB, OSCST dx, WT, MNG, MB/PNET infratentorial, seminoma	c.4403_4406delCTCT c.1716delT c.1196_1197dupAG c.3505delT c.1966C>T p.R656X c.2268_2271delTTTG c.3665delT c.2040+1 G>C c.3726C>A p.Y1242X c.5465A>T p.D1822V c.3583_3584delGA c.3288_3289insTTTC c.328_338dupGTGTCAGCTGT c.5122_5128delGGAGATG c.1966C>T_p.R656X c.3793delA c.2988N/A2_2988N/A1delAGinsCT c.1153delC c.4740G>T p.Q1580H	11 No 6 Yes 2 N/A	16 Alive 2 N/A 1 Dead
Rio Frio et al. [28]	2011	4 F 1 M	16 18 18 15 N/A	5MNG, 3SLCT, 1PPB	c.876_879delAAAG C.2457C>G c.5018_5021delTCAA c.2516C>T c.2805N/A1G>T	3 Yes 2 N/A	4 Alive 1 Dead
Foulkes et al. [32]	2011	5 F 2 M	9 4 2 1.3 5 14 11	SLCT, MNG	c.4050+1delG c.912_919dupAGACTGTC c.1306dupT c.1966C>T c.2117N/A1G>A c.3611_3616delACTACAinsT c.3907_3908delCT	7 Yes	4 Alive 3 N/A
Bahubeshi et al. [33]	2012	F	2.1	CN, PPB	c.4309_4312delGACT	N/A	N/A
Doros et al. [34]	2012	F M	4 0.1	PPB, cERMS, ERMS	c.4309_4312delGACT c.2247C>A	N/A	N/A
Sabbaghian et al. [35]	2012	M	<3	PB	c.1128_1132delAGTAA	N/A	N/A
de Kock et al. [36]	2013	F	3	PPB	c.4555delG	N/A	N/A
Sabbaghian et al. [37]	2013	F	6	SLCT, MNG	c.3270N/A6_4051—1280delinsG	Yes	Alive
Darrat et al. [38]	2013	F	12	MNG	c.1525C>T	Yes	Alive
Stewart et al. [25]	2014	4 F 4 M	2.9 1.8 3.4 4.6 6 3.4 13 17	8PPB, 8NCMH, 3SLCT, DTC	c.2863delA; p.T955fs c.5437G>A; p.Glu1813Lys c.2040+1G>T c.4407_4410delTTCT; p.L1469fs c.1376+1G>T c.5394delA c.3726C>A; P.Y1242X P. Y749X	N/A	N/A
de Kock et al. [16]	2014	5 F 4 M	27 2.8 2.11 17 1.11 3 7 N/A 1.9	3PPB, NCMH, 2MNG, DTC, 6PitB	c.3019C>T; P.Q1007X c.2379T>G c.3505dupT c.2026C>T c.1284delGA c.2379T>G c.3277_3280delAACT c.4309_4312delGACT c.5125G>C	1 No 8 N/A	8 Alive 1 Dead
Rath et al. [39]	2014	M	2.8	CN, PPB, MNG	c.5221_5232delAACAACACCATC	Yes	Alive
Rossing et al. [40]	2014	F	13	MNG, SLCT	c.3647C>A; c.3649T	Yes	Alive
Sahakit Rungruang et al. [41]	2014	F	1	PitB	c.3046delA	N/A	N/A
Palculict et al. [42]	2015	F	3.5	WT	c.2399delG; p.R800fsX5	Yes	Alive
Oost et al. [43]	2015	F	16	SLCT	c.1532_1533delAT	Yes	N/A
de Kock et al. [44]	2015	F	6	cERMS, CN, MNG	c.1196_1197dupAG	N/A	N/A
Schultz et al. [45]	2016	M	5	PPB, MNG, SLCT	c.5096N/A12 G>A + c.5126A>T; p.Asp1709Val	No	Alive
Kuhlen et al. [46]	2016	M	11	PPB, MNG	c.4616C>T	Yes	Alive
Canfarotta et al. [47]	2016	F	7	SLCT, MNG	c.325C>T	No	Alive
de Kock et al. [48]	2016	F	14	MNG, SLCT	c.3540C>A	Yes	Alive
Wu et al. [15,49,50,51]	2014, 2016	3 F	0.9 0.7 12	CN, 2ASK, MNG, OSCT	c.2062C>T c.2450delC c.3540C>G	N/A	2 Alive 1 N/A
Fremerey et al. [52]	2016	F	0.6	CBME, ERMS, CN, PPB	c.3405dupA	Yes	N/A
Bardón-Cancho et al. [14]	2016	F	2	CN, PPB	c.3540C>G	Yes	Alive
Caruso et al. [53]	2017	2 M	34 32	2Liver Cancer	2c.2455T>C	2 Yes	2 Alive
Fernández-Martinez et al. [54]	2017	F	0.11	CN, PPB	c.5387C>T	Yes	Alive
Saskin et al. [55]	2017	F	7	MNG, CN	c.4566_4579dupCTTTG	N/A	Alive
Yoshida et al. [56]	2017	F	12	MNG, DTC	c.5426_5442delGGGATATTTTTGAGTCGinsCA	N/A	Alive
Apellaniz et al. [57]	2019	2 M	2.2 0.9	2Liver Cancer, MNG, PPB, CN, polyps	c.4007delC, p.P1336Lfs11; c.5125G→A, p.D1709N; e c.5113G→C, p.E1705Q	1 Yes 1 No	2 Alive
Bailey et al. [24]	2019	4 M	4 19 3 5	2PPB, RMS, PB	c.5364+1G>A c.1069G>A c.745G>A c.4754C>T	N/A	N/A
Dural et al. [58]	2019	F	10	CN, ERMS	c.5113G>A	N/A	Alive
Haley et al. [29]	2019	F	38	SLCT, MNG	N/A	Yes	Dead
Herriges et al. [59]	2019	M	6	S. Glow, Macrocephaly	N/A	N/A	N/A
Nagasaki et al. [60]	2020	F	6	MNG, DTC	c.4509C>G, p.Y1503X	Yes	Alive
Warren et al. [30]	2020	2 F	5 16	2OSCST	c.2223_2230del; P.S742T16 c.3682C>T;p.Q1228	1 Yes 1 No	1 Alive 1 Dead
Tutor et al. [61]	2020	F	16	PPB, MNG	p.Asp940Ter	N/A	Alive
Apellaniz et al. [62]	2020	M	0.10	ERMS, CN	N/A	N/A	N/A
Zhang et al. [63]	2020	F	16	MNG, ERMS	c.3937delG	N/A	Alive
Chernock et al. [64]	2020	M	17	DTC	c.735N/A8T>G (41/69)	No	Alive
McCluggage et al. [22]	2020	3 F	60 13 14	3ERMS	c.5428G>C, p.D1810H c.5315_5316del; p.F1772fs c.2805N/A2_2817deletionAGATATCGCAATTTT	2 No 1 N/A	N/A
Zhang et al. [65]	2020	F	0.2	PPB	c.C3675A (p.Y1225X)	Yes	N/A
Chong et al. [66]	2021	F	4	WT, MNG, PitB	c.890dupT, p.Leu297Phefs23	Yes	Alive
See et al. [67]	2021	M	12	PPB, Liver CancerPPB like	c.4458dup: p.Ser1487Ilefs5	N/A	Alive
Azzollini et al. [26]	2021	F 2 M	10 0.8 22	SLCT, MNG, PPB, RMS	c.1630C>T p.(Arg544)	3 Yes	3 Alive
Venger et al. [17]	2021	2 F 1 M	0.11 16 3	Macrosomia, 3macrocephaly, WT, PPB, S. Glow, SLCT, MNG	c.4031C>T; p.(Ser1344Leu) c.3073G>T; p.(Glu1025) c.3234_3237dupTGGC	1 No 2 Yes	3 Alive
Juhlin et al. [68]	2021	F	13	DTC	c.2830C>T, p. Arg944Ter (G)	Yes	N/A
Miyama et al. [31]	2021	M	26	RMS	c.5125G>A	N/A	Dead
Rossi et al. [69]	2021	2 M	22 2	2RMS	N/A	N/A	2 Alive
Kaspar et al. [18]	2021	M	1.3	CN, macrosomia, Macrocephaly, PPB	N/A	N/A	N/A
Ni et al. [70]	2021	F	2	CN, SLCT, MNG	c.1088_1089delCTinsAA p.F363X)	N/A	N/A
Darbinyan et al. [23]	2021	7 F	14 12 17 35 37 58 65	7MNG 1ERMS	2 c.5428G>T c.5113G>T c.5428G>T c.5126A>G 2 c.5113G>A	7 Yes	7N/A
Karnak et al. [71]	2021	F	5.1	SLCT, DTC, cERMS, PPB	c.3377delC, c.71delC	N/A	N/A
Gupte et al. [72]	2022	F	13	PB, CBME	c.1847T>A	Yes	Alive
Wilson et al. [73]	2022	F	0.6	ERMS	N/A	N/A	Alive
Pontén et al. [19]	2022	M	1.3	Macrocephaly, WT, S. Glow, PPB, CN	c.4031C>T	Yes	Alive
Hammad et al. [20]	2022	F	0.1	Spinal amarthroma	c.3118_3119insCA	Yes	N/A
Garcia et al. [74]	2022	F	2.6	SLCT	N/A	N/A	N/A
Lau et al. [75]	2022	F	4	SLCT	N/A	N/A	N/A

Sarcoma of the kidney (ASK); medulloepithelioma of the ciliary body (CBME); cystic nephroblastoma (CN); differentiated thyroid cancers (DTC); multinodular goiter (MNG); nasal chondromesenchymal hamartoma (NCMH); pleuropulmonary blastoma (PPB); primitive neuroectodermal tumor (PNET); rhabdomiosarcoma (RMS); glow syndrome (S.GLOW); Sertoli-Leydig cell tumor (SLCT); Wilms tumor (WT).

**Table 3 cancers-15-03681-t003:** Comparison of anthropometric and clinical factors, between published cases and cases detected in Italy. Statistics: percentage or mean (sd).

Factor	Our Study (n = 7)	Literature Review (n = 132)	*p*-Value
Males	4 (57.1)	45 (37.5 out of 120 cases)	0.575
Age first surgery (years)	1.7 (1.4)	10 (7 out of 120 cases)	0.002
Familial	6 (85.7)	55 (70.5 out of 78 cases)	0.676
Death	1 (14.29)	6 (7.1 out of 84 cases)	0.959
PPB	4 (57.1)	54 (40.7)	0.642
CN	3 (42.86)	19 (15.91)	0.186
ASK	1 (14.29)	3 (1.52)	0.351
MNG	3 (28.57)	46 (34.09)	0.912
DTC	1 (14.29)	7 (5.34)	0.874
CBME	1 (14.29)	3 (2.27)	0.488
S.GLOW	1 (14.29)	3 (2.27)	0.488
RMS	1 (14.29)	22 (19.7)	0.721

Sarcoma of the kidney (ASK); medulloepithelioma of the ciliary body (CBME); cystic nephroblastoma (CN); differentiated thyroid cancers (DTC); multinodular goiter (MNG); pleuropulmonary blastoma (PPB); rhabdomiosarcoma (RMS); glow syndrome (S.GLOW).

## Data Availability

The data presented in this study are available in this article.

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
