# Peer review of "DICER1 Syndrome: A Multicenter Surgical Experience and Systematic Review"

_cancers, 2023, doi:10.3390/cancers15143681_

Round 1

Reviewer 1 Report (Previous Reviewer 2)

Thank you for attending to all the reviewers' suggestions.

- Please review the manuscript for grammar once again for example Line 148 '... the patient was reported to have asymmetrical lower limbs... " not as the authors have it currently.

- In the methodology the authors have not completely addressed the issue of their literature search. A literature search methodology should include search engines and search terms.

- As in the presvious review the reviewers suggested that single paragraph writing is difficult to read yet the methodology and discussion section has the same problem. 

- Figure 1: Heading should read "Map of international cases" and the subscipt should be "Map of the international distribution of DICER1 syndrome cases"

It is not the responsibility of reviewers to proof read the manuscript. There are more grammatical errors excluding the examples fore mentioned.

- Table 2: Please change all "case1" etc. to "Case 1" thus with a capitol letter and a space between the numeral and word. This is true for the discussion section.

These are basic editing that is the responsibility of the authors not reviewers. These are not the only examples. Please review your mauniscript.

- Please check the consistancy of your manuscript for example in table 1 all the headings are in lower case, but in table 2 all the headings are in upper case.

- Now that figure 2 is correctly plotted a very important fact is clear. The reviewers suggested to the age related cancer incidence... can the authors please evaluate again the wealth of information their results are showing and integrate it into the manuscript.

There is still grammar and editting concerns that should be attended to.

Author Response

Reviewer 1

Review Report Form

Open Review

Quality of English Language

( ) I am not qualified to assess the quality of English in this paper
( ) English very difficult to understand/incomprehensible
( ) Extensive editing of English language required
( ) Moderate editing of English language required
(x) Minor editing of English language required
( ) English language fine. No issues detected

Yes

Can be improved

Must be improved

Not applicable

Does the introduction provide sufficient background and include all relevant references?

(x)

( )

( )

( )

Are all the cited references relevant to the research?

(x)

( )

( )

( )

Is the research design appropriate?

(x)

( )

( )

( )

Are the methods adequately described?

(x)

( )

( )

( )

Are the results clearly presented?

(x)

( )

( )

( )

Are the conclusions supported by the results?

(x)

( )

( )

( )

Comments and Suggestions for Authors

Thank you for attending to all the reviewers' suggestions.

- Please review the manuscript for grammar once again for example Line 148 '... the patient was reported to have asymmetrical lower limbs... " not as the authors have it currently.

Answer. We thank the Reviewer for the comments.

We altered the manuscript accordingly.

- In the methodology the authors have not completely addressed the issue of their literature search. A literature search methodology should include search engines and search terms.

Answer. We thank the Reviewer for the suggestions.

We added search engines and terms.

- As in the presvious review the reviewers suggested that single paragraph writing is difficult to read yet the methodology and discussion section has the same problem. 

Answer. We thank the Reviewer for the suggestions.

The manuscript is now divided into shorter paragraphs in order to be more readable.

- Figure 1: Heading should read "Map of international cases" and the subscipt should be "Map of the international distribution of DICER1 syndrome cases". It is not the responsibility of reviewers to proof read the manuscript. There are more grammatical errors excluding the examples fore mentioned.

Answer. We thank the Reviewer for the suggestion.

We have changed the heading and subscript. The text has been checked for other grammatical errors

- Table 2: Please change all "case1" etc. to "Case 1" thus with a capitol letter and a space between the numeral and word. This is true for the discussion section. These are basic editing that is the responsibility of the authors not reviewers. These are not the only examples. Please review your mauniscript.

Answer. We thank the Reviewer for the suggestion.

We have done the changes as per suggestions, and we have checked furtherly the text.

- Please check the consistancy of your manuscript for example in table 1 all the headings are in lower case, but in table 2 all the headings are in upper case.

Answer. We thank the Reviewer for the suggestion.

We have done the changes as per suggestions

- Now that figure 2 is correctly plotted a very important fact is clear. The reviewers suggested to the age related cancer incidence... can the authors please evaluate again the wealth of information their results are showing and integrate it into the manuscript.

Answer. We thank you for the comment.

All the information reported into the manuscript have been checked.

Comments on the Quality of English Language

There is still grammar and editting concerns that should be attended to.

Answer. We thank the Reviewer for the comment.

We have checked the text for grammar and editing errors.

Reviewer 2 Report (New Reviewer)

Dear authors,

 You have done a careful review of the DICER1 mutations associated tumours, describing also your own cases. Nevertheless, your great effort doesn’t add new knowledge to the field.

To my point of we your work would highly improve if you discuss aspects the of the molecular pathogenesis of DICER1 mutant tumours. To address these points figures would be highly recommended.

An important issue is to describe if other mutations were found, that could synergistically increase the oncogenic implications of the DICER1 mutations.

The conclusions are, to my point of view, to simple. As suggested before, a more in-depth discussion of the molecular biology of these DICER1 mutant tumours would be very interesting.

Because of all these reasons the manuscript can`t be accepted in the present form.  

Author Response

Reviewer 2

Review Report Form

Open Review

Quality of English Language

(x) I am not qualified to assess the quality of English in this paper
( ) English very difficult to understand/incomprehensible
( ) Extensive editing of English language required
( ) Moderate editing of English language required
( ) Minor editing of English language required
( ) English language fine. No issues detected

Yes

Can be improved

Must be improved

Not applicable

Does the introduction provide sufficient background and include all relevant references?

( )

(x)

( )

( )

Are all the cited references relevant to the research?

( )

(x)

( )

( )

Is the research design appropriate?

( )

( )

(x)

( )

Are the methods adequately described?

(x)

( )

( )

( )

Are the results clearly presented?

(x)

( )

( )

( )

Are the conclusions supported by the results?

( )

(x)

( )

( )

Comments and Suggestions for Authors

Dear authors,

 You have done a careful review of the DICER1 mutations associated tumours, describing also your own cases. Nevertheless, your great effort doesn’t add new knowledge to the field.

To my point of we your work would highly improve if you discuss aspects the of the molecular pathogenesis of DICER1 mutant tumours. To address these points figures would be highly recommended.

An important issue is to describe if other mutations were found, that could synergistically increase the oncogenic implications of the DICER1 mutations.

The conclusions are, to my point of view, to simple. As suggested before, a more in-depth discussion of the molecular biology of these DICER1 mutant tumours would be very interesting.

Because of all these reasons the manuscript can`t be accepted in the present form. 

Answer. We are thankful to the Reviewer suggestion.

Indeed we are planning to move forward within the very next months to deepen our understanding of molecular aspects of the collected patients. Prospectively, we added a paragraph to enhance the manuscript discussion on this topic, citing two papers concerning the biological features of the DICER1 mutations as in - Juhlin CC. On the Chopping Block: Overview of DICER1 Mutations in Endocrine and Neuroendocrine Neoplasms. Surg Pathol Clin. 2023) and - Robertson JC et al. DICER1 Syndrome: DICER1 Mutations in Rare Cancers. Cancers (Basel). 2018) (see lines 284-292):

“DICER1 tumor suppressor gene is a master regulator of miRNA processing and may be associated with somatic mutations or gene deletions of the remaining allele. The DICER1 protein deficit is interfering with miRNA output therefore interfering with post transcriptional gene regulation leading to neoplasm development [84]. Eighty-eight DICER1 mutations were described for PPB; while most of these are in the regions encoding defined domains, such as helicase domains, the Dicer dimerization domain (DDD), the Piwi/Argonaute, Zwille (PAZ) domain, the RNase III domains, and the double-stranded RNA-binding domain; other mutations reside outside these domains [85]”.

  1. Juhlin, C.C. On the Chopping Block: Overview of DICER1 Mutations in Endocrine and Neuroendocrine Neoplasms. Surg Pathol Clin 2023, 16, 107-118.
  2. Robertson, J.C.; Jorcyk, C.L.; Oxford, J.T. DICER1 Syndrome: DICER1 Mutations in Rare Cancers. Cancers (Basel) 2018, 10, 143.

Reviewer 3 Report (New Reviewer)

No comment. 

Author Response

Reviewer 3

Review Report Form

Open Review

Quality of English Language

(x) I am not qualified to assess the quality of English in this paper
( ) English very difficult to understand/incomprehensible
( ) Extensive editing of English language required
( ) Moderate editing of English language required
( ) Minor editing of English language required
( ) English language fine. No issues detected

Yes

Can be improved

Must be improved

Not applicable

Does the introduction provide sufficient background and include all relevant references?

( )

(x)

( )

( )

Are all the cited references relevant to the research?

(x)

( )

( )

( )

Is the research design appropriate?

( )

(x)

( )

( )

Are the methods adequately described?

( )

( )

( )

(x)

Are the results clearly presented?

( )

( )

(x)

( )

Are the conclusions supported by the results?

( )

( )

(x)

( )

Comments and Suggestions for Authors

No comment. 

Answer. We thank the Reviewer.

Round 2

Reviewer 1 Report (Previous Reviewer 2)

I am happy with the changes.

Author Response

Reviewer 1

Review Report Form

Open Review

( ) I would not like to sign my review report

(x) I would like to sign my review report

Quality of English Language

( ) I am not qualified to assess the quality of English in this paper

( ) English very difficult to understand/incomprehensible

( ) Extensive editing of English language required

( ) Moderate editing of English language required

( ) Minor editing of English language required

(x) English language fine. No issues detected

Comments and Suggestions for Authors

I am happy with the changes.

Answer. We thank the Reviewer for the comment.

Reviewer 2 Report (New Reviewer)

Dear authors,

 You have improved the manuscript by adding some aspects of the molecular pathogenesis of DICER 1 mutant tumours to the discussion. Nevertheless, a better explanation is needed of how micro-RNAs regulate genes’ expressions and how an alteration in their function can induce tumour growth. As suggested in my previous review, images of this topic are still missing. 

In the discussion is also being said: “DICER1 guarantees the correct expression of numerous neurocortical genes and contributes to the correct neuronal transmission, suggesting not only an oncogenic role but also a neuro-signalling function.” This neuro-signalling function of DICER1 gene must be better explained

Finally, under the conclusions is stated: “Its manifestations can be of neoplastic or non-neoplastic nature…”. Which are these non-neoplastic conditions? Or do the authors mean malignant and non-malignant?

Author Response

Reviewer 2

Review Report Form

Open Review

Quality of English Language

(x) I am not qualified to assess the quality of English in this paper
( ) English very difficult to understand/incomprehensible
( ) Extensive editing of English language required
( ) Moderate editing of English language required
( ) Minor editing of English language required
( ) English language fine. No issues detected

Yes

Can be improved

Must be improved

Not applicable

Does the introduction provide sufficient background and include all relevant references?

( )

( )

(x)

( )

Are all the cited references relevant to the research?

(x)

( )

( )

( )

Is the research design appropriate?

(x)

( )

( )

( )

Are the methods adequately described?

(x)

( )

( )

( )

Are the results clearly presented?

(x)

( )

( )

( )

Are the conclusions supported by the results?

( )

( )

(x)

( )

Comments and Suggestions for Authors

Dear authors,

 You have improved the manuscript by adding some aspects of the molecular pathogenesis of DICER 1 mutant tumours to the discussion. Nevertheless, a better explanation is needed of how micro-RNAs regulate genes’ expressions and how an alteration in their function can induce tumour growth. As suggested in my previous review, images of this topic are still missing. 

Answer. We thank the Reviewer for the suggestion.

We have changed the Introduction section (see lines 68-89):

“The DICER1 gene is located on chromosome 14q32.13 and encodes for a protein of ap-proximately 200kDa: it is an endoribonuclease of the RNase III family involved in the production and maturation process of most microRNAs (miRNAs), which are ~22-nucleotide small noncoding RNAs, known to have a key role in the post-transcriptional regulation of mRNA [5,7]. The deregulation of miRNAs has a pro-oncogenic effect: the overexpression of one miRNA can act to inhibit the protein translation of a tumor suppressor gene, while the downregulation of another miRNA can increase the protein level of an oncogene [8-12]. miRNAs are generated from the nascent primary miRNA (pri-miRNA) transcripts through two consecutive cleavage events. The pri-miRNA is processed by DROSHA in the nucleus, which releases a hairpin-shaped precursor (pre-miRNA). Pre-miRNA move from the nucleus to cytoplasm by exportin 5 (XPO5) and are cleaved by DICER. The resulting small RNA duplex is loaded onto the Argonaute (AGO) protein, that retains only one strand of mature miRNA and removes the other one. The miRNA-loaded AGO associates with other cofactors, and constitutes the effector complex called the RNA-induced silencing complex (RISC). The miRISC induces the decay of mRNA and translational suppression, interacting with the com-plementary sequences in the 3′-untranslated region (3′-UTR) of target gene mRNA. The miRNA target most mRNA, and in this way they have important regulatory roles in different physiological and developmental processes. Overall, miRNA-mediated gene expression control is critical for cell response to oxidative stress, hypoxia, and DNA damage, and for this reason it is involved in human diseases, such as cancer [13]”.

  1. Ali Syeda, Z.; Langden, S.S.S.; Munkhzul, C.; Lee, M.; Song, S.J. Regulatory Mechanism of MicroRNA Expression in Cancer. Int J Mol Sci 2020, 21, 1723, doi: 10.3390/ijms21051723.

Moreover, a schematic illustration has been added in the text.

In the discussion is also being said: “DICER1 guarantees the correct expression of numerous neurocortical genes and contributes to the correct neuronal transmission, suggesting not only an oncogenic role but also a neuro-signalling function.” This neuro-signalling function of DICER1 gene must be better explained

Answer. We thank the Reviewer for the comment.

The sentence has been changed to (see lines 288-291):

“DICER1 guarantees the correct expression of numerous neurocortical genes and con-tributes to the correct neuronal transmission, suggesting a role not only in the oncogenesis, but also in the nervous system development [5,79,82]”.

See for example, DICER1-associated macrocephaly.

Finally, under the conclusions is stated: “Its manifestations can be of neoplastic or non-neoplastic nature…”. Which are these non-neoplastic conditions? Or do the authors mean malignant and non-malignant?

Answer. We thank the Reviewer for the comment.

The sentence has been changed to “Its manifestations can be of malignant or non-malignant nature and can occur exclusively or combined in each patient in a different way” (see lines 343-344).

Reviewer 3 Report (New Reviewer)

The manuscript has been improved to satisfy the reviewer. 

Author Response

Reviewer 3

Review Report Form

Open Review

Quality of English Language

(x) I am not qualified to assess the quality of English in this paper
( ) English very difficult to understand/incomprehensible
( ) Extensive editing of English language required
( ) Moderate editing of English language required
( ) Minor editing of English language required
( ) English language fine. No issues detected

Yes

Can be improved

Must be improved

Not applicable

Does the introduction provide sufficient background and include all relevant references?

( )

(x)

( )

( )

Are all the cited references relevant to the research?

(x)

( )

( )

( )

Is the research design appropriate?

(x)

( )

( )

( )

Are the methods adequately described?

(x)

( )

( )

( )

Are the results clearly presented?

(x)

( )

( )

( )

Are the conclusions supported by the results?

(x)

( )

( )

( )

Comments and Suggestions for Authors

The manuscript has been improved to satisfy the reviewer. 

Answer. We thank the Reviewer for the comment.

This manuscript is a resubmission of an earlier submission. The following is a list of the peer review reports and author responses from that submission.

Round 1

Reviewer 1 Report

In this paper the authors report on 7 children with DICER1 syndrome and compare these children with a literature cohort after extensive reviewing of literature.

I have several concerns regarding this manuscript. The authors state that they describe "A nationwide report of DICER in the Italian pediatric population". However, they do not explain how these patients were selected. Which patients were offered NGS multi-gene panel? Seven patients seems a low number for a nationwide cohort and likely data collection was biased by using surgical registries. 

Because the number of patients is very low there is no new message in this report. The authors make comparissons with the cohort derived from literature and attribute percentages to the findings in either cohort. However, there own cohort is much too small for relevant calculations and it does not make sense to compare a paediatric cohort with a cohort that also includes many adult patients since DICER1 syndrome is characterized by age specific cancer penetrance. 

The methods of the extensive literature search are not mentioned. 

Reviewer 2 Report

The manuscrip is of importance to not only the oncology, but paediatric community.

Language and grammar:

- The language and grammar should be reviewed please, consider the following (but not only these):

-         -  Replace “nowadays” in line 30 (Simple summary), line 40 (Abstract), line 75 (Introduction) and line 292 (Conclusions) with “currently” or “presently”

-         -  The word “both” may be deleted in lines 35 (Simple summary), line 49 (Abstract) and line 102 (Materials and Methods)

-        -   Replace line 68 “… there are no significant gender differences, neither concerning ethnic groups.” With “… there are no significant differences in gender nor ethnic groups.”

-          - Line 76: Replace “outcome” with “outcomes”.

-          - Lines 97 – 98: please add the word “years” after 8 and 12.

-        -  Line 99: replace “should have been” with “was”

-        -   In the results section please place a space between numerals and letters i.e. 12 years and 40 days. Also consistently use either the word “years” or the abbreviation.

-       -   In the results section please be reminded that you are describing infants. Thus a statement like “The patient reported…” (line 144) is illogical. A child of two years can’t tell you that he/she has asymmetrical lower limbs. Please review other such statements.

-         - The same argument applies to “sex” and “gender”. “Sex” is an anatomical described allocation, where “gender” is a decision a person of emotional maturity makes. Thus none of these patients could identify their genders. The reviewer suggests the word “sex”.

-         - In the discussion (line 206) replace “muticentric” with “multicenter”.

-          - Line 212 “bare not “bear”

-          - Replace “silent clinical history” with “silent clinical phenotype” (line 213)

-         -  Line 225 “familiarity” is the incorrect word.

There are numerous word in singular form that should be plural.

Content:

-        - The authors make a point of highlighting delayed diagnoses of DICER-1 in relation to surgery. The reason for this is not clear. The diagnosis may delayed in any medical situation, even in oncology. Are the authors referring to diagnosis i.e. biopsy?

-         - Regarding the follow up program (line 94): “…in the first months…” may you please specify if you mean as a “baseline” or “monthly”. If “monthly” please indicate the number of months this should be done monthly.

-         - The authors may already define pleuropulmonary blastoma (PPB) in line 58 rather than line 99.

-         - Nowhere at the beginning of the results section do the authors refer to “Table 1”. Please indicate this in the opening line already not in line 179. The same applies for the reference to “Table 2”. Note it in line 174 already. Same for “Table 3”.

-         - The single paragraph results section is extremely difficult to read. The reviewer suggests reframing the results section in smaller paragraphs.

-         - Please make a clear paragraph break between the institutional results and the literature results (line 174)

-        -  In the content and the tables the word “Familiarity” should be changed to “Familial”. “Familiarity” means something is well known to someone.

-        - In line 226 it isn’t only the knowledge that is lacking but “awareness”. Please include the word “awareness”.

Figure 2:

-          From an academic perspective it is advised that in Figure 2 the y-axis is the number of patients and the x-axis the ages. Normally the determining denominator (age in this case) is on the x-axis. Currently the graph doesn’t make sense.

Tables:

-         - In the content and the tables the word “Familiarity” should be changed to “Familial”. “Familiarity” means something is well known to someone.

-         -  In the column “Follow up” please replace with “Outcome status”

The conclusion:

- The conclusion is too long. Most of the points may be best placed at the end of the discussion as recommendations also since this manuscript highlights the awareness towards DICER-1.

- A conclusion is a short highlight of the most important messages coming out of the study.